# Matching Drug Metabolites from Non-Targeted Metabolomics to Self-Reported Medication in the Qatar Biobank Study

**DOI:** 10.3390/metabo12030249

**Published:** 2022-03-16

**Authors:** Karsten Suhre, Nisha Stephan, Shaza Zaghlool, Chris R. Triggle, Richard J. Robinson, Anne M. Evans, Anna Halama

**Affiliations:** 1Bioinformatics Core, Weill Cornell Medicine-Qatar, Education City, Doha 24144, Qatar; nis2034@qatar-med.cornell.edu (N.S.); sbz2002@qatar-med.cornell.edu (S.Z.); amh2025@qatar-med.cornell.edu (A.H.); 2Department of Physiology and Biophysics, Weill Cornell Medicine, New York, NY 10065, USA; 3Departments of Medical Education and Pharmacology, Weill Cornell Medicine-Qatar, Education City, Doha 24144, Qatar; cht2011@qatar-med.cornell.edu; 4Metabolon Inc., Morrisville, NC 27560, USA; rrobinson@metabolon.com (R.J.R.); aevans@metabolon.com (A.M.E.)

**Keywords:** medication, drug metabolites, population studies, non-targeted metabolomics

## Abstract

Modern metabolomics platforms are able to identify many drug-related metabolites in blood samples. Applied to population-based biobank studies, the detection of drug metabolites can then be used as a proxy for medication use or serve as a validation tool for questionnaire-based health assessments. However, it is not clear how well detection of drug metabolites in blood samples matches information on self-reported medication provided by study participants. Here, we curate free-text responses to a drug-usage questionnaire from 6000 participants of the Qatar Biobank (QBB) using standardized WHO Anatomical Therapeutic Chemical (ATC) Classification System codes and compare the occurrence of these ATC terms to the detection of drug-related metabolites in matching blood plasma samples from 2807 QBB participants for which we collected non-targeted metabolomics data. We found that the detection of 22 drug-related metabolites significantly associated with the self-reported use of the corresponding medication. Good agreement of self-reported medication with non-targeted metabolomics was observed, with self-reported drugs and their metabolites being detected in a same blood sample in 79.4% of the cases. On the other hand, only 29.5% of detected drug metabolites matched to self-reported medication. Possible explanations for differences include under-reporting of over-the-counter medications from the study participants, such as paracetamol, misannotation of low abundance metabolites, such as metformin, and inability of the current methods to detect them. Taken together, our study provides a broad real-world view of what to expect from large non-targeted metabolomics measurements in population-based biobank studies and indicates areas where further improvements can be made.

## 1. Introduction

Population-based and epidemiological biobank studies provide essential information about the health of the general population and are key resources for research into disease etiology and comorbidities [1,2,3,4]. There are numerous large biobank projects around the globe, including in Europe, North America, Australia, China, Japan, Korea and Qatar [5,6,7,8,9] that gather biological samples, clinical data, basic laboratory test results, along with imaging data and questionaries. These data sets are primarily obtained from volunteers without any specific health-related inclusion/exclusion criteria [10]. Further insights into disease etiologies and comorbidities are derived from these samples as a result of technological advancement and the introduction of the -omics sciences (genomics, transcriptomics, proteomics, metabolomics), along with large electronic databases that are capable of storing and managing the large data sets associated with the subject information into widely accessible population-based biobanks [11,12].

Statistics on the use of prescription drugs and over-the-counter medication reflect actual patient behavior in disease management. They can even provide insights into the genetic bases of complex diseases, as shown in a recent genome-wide association study on medication use in UK Biobank [13]. However, information on medication use, diet and lifestyle is generally obtained from questionnaires, which may be inaccurate due to multiple factors, including communication issues, unclear wording or too extensive questionaries [14]. Such biases may substantially affect the analysis of the actual drug exposure which can result in false estimation of used medication and their effects [15,16]. A previous study, assessing the concordance between the information on medication use derived in parallel from questionnaires and from pharmacy database records reported good concordance for medication used to treat chronic disorders, such as cardiovascular disease, type 2 diabetes (T2D) and hypothyroidism, but poor concordance for medication used over shorter periods of time [17]. Therefore, validation of questionnaire-derived medication data is critical for accurate analysis.

In this context, non-targeted metabolomics, designed for the broad characterization of ideally all relevant small molecules in a biological sample, can help to answer the question of how well self-reported and blood-detected drug uses match. Indeed, a pilot study conducted on 83 subjects deployed metabolomics to test whether questionnaire-derived medication use could be verified using metabolomics readouts from urine [18]. The study showed that molecular evidence for many classes of medication could be obtained from urine metabolic profiles. Nevertheless, some of the drugs, predominantly those extensively metabolized and excreted by the liver (omeprazole, rabeprazole, atorvastatin, and simvastatin), were not captured by the metabolomics analyses of urine. However, the potential of untargeted metabolic profiling applied in the plasma samples as a strategy for the verification of the questionary-derived data on the medication usage has not been previously explored.

Here, we analyze self-reported drug use from 6000 participants of the Qatar Biobank (QBB) [9,19] together with non-targeted metabolomics measurements made on 2807 matching blood plasma samples. We curate and annotate the self-reported drug use data using ATC terms and ask how self-reported QBB drug use compares with top self-reported drug use in the UK Biobank (UKB). We then analyze the metabolomics data to ask (1) how well self-reported drug use is reflected in blood detected metabolites, (2) which drugs are potentially under-reported by study participants, that is, only detected in blood, and (3) which drugs are not detected using the currently available metabolomics platforms, that is, only self-reported by the participants? We discuss relevant examples in more detail, including the use of paracetamol, metformin, statins, and psychoactive drugs.

## 2. Results

### 2.1. Demographics

Qatar Biobank is a population-based cohort study in Qatar [9,19]. In this analysis, we had access to data obtained from 6000 subjects of Qatari nationality. Overall, the analyzed cohort is relatively young compared with other biobanks, with an average participant age of ~40 years (Table 1). Most of the participants were aged between 25 and 54 years and only ~2% of participants were older than 65 years (Figure 1a). A total of 56% of the study participants were female. The average BMI was 28.9 kg/m^2^ and the average HbA1c was 5.7%. In total, 17% of the subjects self-reported to have been diagnosed with diabetes, 30% of the participants reported to be treated for high cholesterol level, 16% for high blood pressure, and 10% reported to be regular smokers.

### 2.2. Self-Reported Medication in QBB

Free-text responses of 6000 study participants of QBB to the question “Are you taking any over-the-counter medication or prescription medicines regularly? For example, daily, weekly, monthly or every few months—such as depot injections?” were analyzed. In total, 3086 participants (51.4%) provided at least one affirmative answer. The frequency of medication use increased linearly from ~25% at age 20 years to ~100% at age 70 years, with females reporting more medication use than men (Figure 1b). Space for up to 30 free-text answers was provided by the electronic questionnaire. The largest number of registered entries was seventeen, indicating that likely no medication use was omitted by participants due to lack of space to respond. In total, 7757 individual entries were recorded, covering 3450 distinct text items. The responses provided by the participants were in free-text format and varied in detail, ranging from “unspecified medication for disease X” to “specific drug brand, dose, and frequency of usage”. The responses occasionally contained spelling errors or abbreviations. Information on dosage and frequency of medication use were only provided sparsely and non-systematically and were, therefore, not further included in the annotation and analysis.

In a first step, all unique entries were manually annotated with an active molecule name. Where possible, these were then matched to a Drugbank [20,21] entry (e.g., “*Aspirin*” -> “*acetylsalicylic acid*” -> Drugbank identifier “*DB00945*”). Spelling errors were corrected where possible, leaving only 75 of the 3450 answer codes (2.2%) as “*not annotated*”. The most frequently reported drugs among the 7757 individual response items were metformin (N = 533), levothyroxine (N = 492), acetylsalicylic acid (aspirin) (N = 309), atorvastatin (N = 234), and insulin (N = 224). The most frequently mentioned supplements were vitamin D (N = 1029), multivitamin preparations (N = 462), iron (N = 223), calcium (N = 169), omega-3 fatty acids (N = 165) and vitamin B (N = 155). In cases where only a general indication for the drug was provided, the active molecule was annotated as “*unspecified [indication]*” (e.g., “*unspecified hypertension*”), leading to 68 unspecified types of indications that cover 1224 of all responses (15.8%). The most frequent unspecified indications were “*unspecified dyslipidemia*” (N = 278), “*unspecified hypertension*” (N = 259) and “*unspecified diabetes*” (N = 153).

Links to the WHO Anatomical Therapeutic Chemical (ATC) Classification System provided by Drugbank were then used to assign ATC codes to all entries that had a Drugbank identifier. Note that an active molecule can have multiple ATC codes assigned [22]. Drugs that are composed of multiple active molecules were split into multiple entries and treated as distinct items (e.g., “*Exforge*” -> “*amlodipine and valsartan*”). In total, 433 unique molecule names were annotated. Of these, 298 molecules had a Drugbank identifier, and 272 of these also had an ATC code (Table 2 and Figure 2). Appendix A provides individual counts for all annotated molecules and ATC codes.

Using the ATC coding system, we then analyzed the use of specific drug classes by age and sex (Figure 3). The most frequently reported medication in QBB were “*drugs used in diabetes*” (A10) and “*lipid modifying agents*” (C10). Over 40% of individuals aged 55 and above reported taking either or both types of these drugs. More female participants used “*drugs used in diabetes*”, whereas more males took “*lipid modifying agents*”. “*Thyroid therapy*” (H03) was predominant in females, reported by some subjects already at a young age of 18–24 years (~5%) and its usage increased with the age with 20% of females reported using such drugs at the age of 55 years and above. “*Vitamins*” (A11) and “*antianemic preparations*” (B03) were also predominantly reported by female study participants.

### 2.3. Comparison of Self-Reported Drug Use in QBB and UK Biobank 

To evaluate the consistency of self-reported medication in QBB with what is observed in other population studies, we used UK Biobank data. The average age of the UKB participants at recruitment (56.5 yrs, s.d. = 8 yrs) was fifteen years higher than that of the QBB participant, whereas the average BMI (27.3, s.d. = 4.8) and the proportion of female participants (54.4%) in UKB was comparable to QBB. Despite their older age, only 5.3% of the UKB participants were diagnosed with diabetes, compared with 17.4% in QBB. Medication use in UKB was obtained via a verbal interview with a trained nurse (see methods). 372,854 out of 500,000 participants (74.6%) in UKB reported taking at least one medication item (Appendix A), which is substantially higher than in QBB (51.4%) but nevertheless in agreement with the observed increase in medication use with age (Figure 1b).

The most frequently reported items in UKB were paracetamol (N = 102,058), aspirin (N = 72,926), ibuprofen (N = 67,388), simvastatin (N = 64,538), omeprazole (N = 35,724), bendroflumethiazide (N = 30,052), ramipril (N = 27,363), amlodipine (N = 26,198), levothyroxine (N = 24,081), and atorvastatin (N = 21,516). Among the dietary supplements the most frequently reported items in UKB were glucosamine products (N = 34,219), cod liver oil capsules (N = 29,961), omega-3/fish oil supplements (N = 19,877), and multivitamin supplements (N = 16,504). It is noteworthy that the three most frequently reported medications in UKB were analgesics (paracetamol, aspirin, and ibuprofen), whereas only a small fraction of QBB participants reported their use. Other frequently reported medications in UKB were the anti-dyslipidemia drugs simvastatin and atorvastatin, as well as drugs used to control high blood pressure, including bendroflumethiazide, ramipril, and amlodipine. In contrast to QBB, metformin and other blood glucose controlling agents were not among the most frequently reported drugs in UKB.

Overall, there were substantial differences in self-reported medication use between UKB and QBB. These may in part be due to differences in demographics (older participants taking more medication), lifestyle (e.g., use of vitamin supplements) and disease prevalence (i.e., diabetes rates), but there also appears to be a reporting bias for some of the over-the-counter drugs (e.g., only 28 of the 6000 QBB participants report using paracetamol, whereas 1 in 5 UKB participants do). It may, therefore, be of interest to evaluate the self-reported use of these medications with blood detection, as we shall investigate in the following section.

### 2.4. Linking Drug Metabolites Detected in Blood and Self-Reported Medication

For independent validation of the questionnaire-derived medication use, samples of 3000 QBB participants were analyzed on the non-targeted HD4 metabolomics platform of Metabolon Inc. (Morrisville, NC, USA) that is operated by the Anti-Doping Laboratory–Qatar. For 2807 of these samples, we had access to matching questionnaire data. In total, semi-quantitative levels of 1159 metabolites were reported; of these, 119 were annotated as drugs and assigned by Metabolon to 13 different drug categories (Appendix A). In 80% of the blood samples at least one drug metabolite related to the category “*analgesics/anesthetics*” was detected (see methods), 46.5% of the blood samples contained at least one molecule related to drugs targeting “*metabolic processes*”, and 22.3% of the samples contained at least one metabolite related to “*drugs used to treat cardiovascular problems*”.

We investigated the overlap between self-reported medication use and the detection of drug-related metabolites in the participants’ blood samples in a hypothesis free approach by testing all possible 2 × 2 tables constituted by the counts of “*detected*” (yes/no) versus “*self-reported*” (yes/no) drug molecules for significant deviation from the null using a Fisher exact test. We accounted for multiple testing by the number of tested metabolites (N = 119) times the number of self-reported medication items (N = 394), yielding a conservative Bonferroni level of significance of *p* < 1.1 × 10^−6^ (Appendix A).

In total, 82 metabolite–drug pairs showed a significant overlap between self-reported drug usage and blood-detected metabolites at this level of significance. Multiple metabolites can associate with the same drug (e.g., multiple detected acetaminophen metabolites associated with paracetamol usage) and multiple drugs that are used to treat a single underlying condition (e.g., self-reported metformin use with drugs used to treat diabetes comorbidities). To avoid confounding or counting the same medication multiple times, we applied a stringent mutual-best-hit criterion, meaning that we considered only the single strongest associations for each self-reported medication and for each detected drug metabolite, respectively, and this only if they were their mutual strongest association. We found 22 such mutual-best-hit matches that reached Bonferroni significance (Table 3). In all 22 cases, the self-reported medication and the detected drug metabolite were identical or biochemically related (e.g., self-reported use of atorvastatin associated with detection of o-hydroxyatorvastatin in blood). When including mutual-best-hits up to nominal significance (*p*-value < 0.05), we found 59 self-reported medication-detected drug metabolite pairs (Appendix A), many of which were biochemically related or identical (e.g., ibuprofen, tramadol, verapamil, tamoxifen).

Metabolites of undetermined biochemical identity (unknowns) may correspond to drug metabolites and can potentially be identified by their association with self-reported medication. To identify possible cases, we repeated the analysis by including all unknows with >50% missing values (N = 40) (see Appendix A). All associations between named metabolites and self-reported medication remained unchanged. In total, 6 unknowns had associations that were significant after correcting for 394 self-reported medications times 40 unknowns (*p* < 3.2 × 10^−6^). A total of 5 of these unknowns were associated with “metformin”, “unspecified hypertension”, and “vitamin d” and might be confounded by indication rather than constitute metabolites of the associated medication. One molecule (X − 17348) was associated with esomeprazole (14 out of 26 self-reported esomeprazole users had X − 17348 detected in their blood, whereas X − 17348 was detected in total in only 11.6% of the samples, *p*-value = 1.6 × 10^−7^, Fisher test).

Among the 22 mutual-best-hits that were significant at a conservative Bonferroni level of significance, in all these cases the number of self-reported medications was lower than the number of matching drug metabolites detected in blood. The average fraction of cases in which a matching drug metabolite was both, self-reported and detected in blood was 26.7%. The average fraction of self-reported-only cases was 7.4% and that of detected-only cases was 65.8%. If a drug had been self-reported by a study participant, it was also detected in that participant’s blood sample in 79.4% of the cases. On the other hand, if a drug metabolite had been detected in a blood sample, the corresponding study participant reported taking the matching medication in only 29.5% of the cases. Thus, there is a good recall of self-reported medication using non-targeted metabolomics. On the other hand, a large amount of detection of drug metabolites without a matching self-reported medication also occurred. We investigate relevant examples of these cases in the following section.

### 2.5. Agreement between Questionnaire-Based and Metabolomics-Derived Treatment of Hypertension, Dyslipidemia, and Diabetes

Because questionnaire-derived medication usage data can be prone to biases, we investigated the concordance between drug metabolites detected in blood, the participant self-reported medication, and in addition responses regarding the treatment of frequent disorders with tablets. Note that the latter were addressed in an independent part of the QBB questionnaire and allowed us to cross-validate the consistency of the questionnaire responses within QBB. We focused on hypertension, dyslipidemia, and diabetes because these disorders were reported by a large fraction of the QBB participants.

While investigating the overlap between 255 participants reporting to be treated with tablets for hypertension, 322 participants for whom self-reported drugs were identified as those prescribed for hypertension, and 536 participants in whom the hypertension drug molecules were detected, we found overlap between 21% of subjects (Figure 4a). For around 13% of subjects, a lack in concordance between the response to the question “*Hypertension treated with tablets*” and self-reported medication record was found. In 50% of the cases where a hypertensive drug was detected, there was no positive response to the general question, nor mention of a specific medication. Conversely, 19% of those subjects who reported taking hypertensive medication, either by name or as a general response, showed no detectable level of any associated drug.

The patterns found for dyslipidemia were similar showing a lack of concordance between the two questionnaire-derived (“self-reported medication” versus “treatment with tablets”) items in 19% of the subjects (Figure 3). For 46% of the subjects reporting dyslipidemia drug intake, the drug molecule or one its metabolites were not detected. The patterns found for diabetes were slightly different; for around 70% of the subjects, the signal for an antidiabetic drug was detected but not reported in the questionnaire, and in only 5% of subjects was the intake of diabetes medications reported but not detected (Figure 3).

To further investigate potential under- or over-detections, we focused on the following pertinent examples: (1) the analgesic acetaminophen, (2) psychoactive medications, and (3) the diabetes drug metformin.

### 2.6. Paracetamol and Psychoactive Drugs as Examples of Under-Reported Medications

Acetaminophen, available under the brand name paracetamol, is a frequently used analgesic drug that is sold over the counter without prescription. Only 17 out of 2807 QBB participants (or 28 out of 6000) self-reported taking this drug, whereas over one fifth of the UKB participants reported using it (Appendix A). Nine acetaminophen metabolites have been detected jointly in blood samples of 310 QBB participants (Table 4 & Figure 5). These nine molecules were detected in two of Metabolon’s platforms, eight were measured in LC/MS negative mode and one in LC/MS positive mode. The chromatographic retention characteristics and the molecular masses differ widely across the detected molecules. Taken together, these observations suggest that these are true positive detections, and that acetaminophen is likely a drug that has been under-reported, where at least 310 QBB participants were taking the drug.

INTERESTINGLY, the non-targeted metabolomics data covers the complete acetaminophen metabolic pathway [23]. Acetaminophen can be eliminated from the human body via three general detoxification pathways: (1) sulfation involving SULT gene products, (2) glucuronidation by UGT gene products, and (3) reduction by glutathione using CYP gene products. We detected the active compound (4-acetamidophenol) and eight of its metabolites that together involve molecules related to all three of these detoxification pathways (Table 4 and Figure 4), that is, 4-acetaminophen sulfate, 2-hydroxyacetaminophen sulfate, and 2-methoxyacetaminophen sulfate related to sulfation (SULT), 2-methoxyacetaminophen glucuronide and 4-acetamidophenyl glucuronide for glucuronidation (UGT), and 3-(N-acetyl-L-cystein-S-yl) acetaminophen and 3-(cystein-S-yl)acetaminophen reflecting the conversion of acetaminophen to reactive intermediates that can then bind to the cysteine thiol of a glutathione molecule (CPY), plus 3-(methylthio)acetaminophen sulfate which is related to two pathways (SULT and CYP) (Figure 4). Interestingly, N-acetyl-cysteine (NAC) has been reported as an effective antidote for acetaminophen overdosing (when administered early). The primary therapeutic effect of NAC is replenishment of glutathione [24]. The detection of 3-(cystein-S-yl)acetaminophen in blood samples from generally healthy study participants suggests that this pathway may also constitute a detoxification route that is taken normally by the human body. As a side-insight to the general objective of this study, this example illustrates how non-targeted metabolomics can map-out complete drug-detoxification pathways, suggesting potential future applications to studies of drug action and pharmacogenomics.

A second class of medication that was detected but appears to be under-reported in the QBB population are psychoactive drugs. Metabolites of the antidepressant citalopram, a selective serotonin reuptake inhibitor (SSRI), were detected in over twenty samples (citalopram propionate*, citalopram/escitalopram, desmethylcitalopram*, N_det_ = 34, 31, 21), but only twelve individuals (0.4%) reported taking citalopram, compared with 1.9% in UKB. Fluoxetine, another SSRI, was detected 14 times, but only reported three times (two of the three self-reports overlap with detections), compared with 1.4% in UKB. A metabolite of fluoxetine metabolism, norfluoxetine was detected 16 times. Unreported but detected anti-depressants include duloxetine (N_det_ = 5) and its metabolites 4-hydroxy duloxetine glucuronide* (N_det_ = 4) and 5-hydroxy-6-methoxy duloxetine sulfate* (N_det_ = 4), midazolam (N_det_ = 4), mirtazapine (N_det_ = 2), and imipramine (N_det_ = 2). The substantially higher self-reporting of psychoactive drugs in UKB suggests that these metabolites are at least in part true positive detections and represent under reported medications in QBB.

### 2.7. Metformin as an Example of a Drug Susceptible to False Positive Detection

Metformin is a frequently used first-line drug used in diabetes care. For 92.3% of participants who reported taking metformin, the molecule was also detected in their blood samples (203 out of 220). However, in 76.0% of the cases where metformin was detected in a blood sample, the study participants did not report taking it (643 out of 846). As metformin is a prescription drug, the likelihood of study participants not reporting it is much lower than in the case of acetaminophen. Some cases of under-reporting may be due to participants only reporting taking “*diabetes medication*” without specifying the actual drug. It has also been reported that individuals at high risk of diabetes (e.g., obesity, high HbA1c) may be taking metformin in Qatar without an actual diabetes diagnosis or prescription. However, although these factors may explain some of the false positive cases, they are unlikely to account for their majority. To investigate this further, we restricted the samples to two groups, one that is very unlikely to take metformin (BMI < 25 kg/m^2^, HbA1c < 5.7%, no diagnosis of diabetes, no self-reported diabetes medication) and one that self-reported taking metformin. We then compared the quantitative metabolomics read-outs for metformin (batch-normalized ion counts) between both groups (Figure 6). The difference of the average log2-scaled metformin levels in blood between both groups was highly significant (*p* < 10^−16^). The much lower levels of metformin found in the group of individuals that most likely did not take metformin suggest that these are due to false positive identifications or experimental artifacts. We investigated a possible carry-over as a source for the false positive samples using blanks and information on successively measured samples using confidential process information from Metabolon, but could not find any significant evidence of this phenomenon.

## 3. Discussion

Our study describes the potential of non-targeted metabolomics in the assessment of questionary-derived data on medication use. In this analysis we found evidence of the under-reporting of drugs within the QBB participants, as it is a likely the case in acetaminophen and psychoactive drug usage. The under-reporting of medication has many potential possible sources. It is possible that cultural biases and norms influence the comfort level of participants disclosing their complete medication list. This is potentially the underlying cause in the drastic but presumed under-reporting of psychoactive medications within the QBB population as compared with UKB. Under-reporting is also likely exacerbated by the prevalence of certain drugs, such as acetaminophen. Acetaminophen is present in many different formulations including common cold and flu medications, which could mean that the participant(s) simply did not know that acetaminophen was present in the over-the-counter medicine they used. It should also be noted that the differences in the reporting of pain-killer usage may be due to differences in the interpretation of the questions posed by QBB and UKB, i.e., what the individual study participants may consider as a “medication”.

In contrast, we also found evidence of false positive detection of select drugs as shown by metformin. There are various technical method-specific issues which can lead to over-detection of drugs in samples during untargeted metabolomic analysis, especially of large sample sets as encountered in this study. Data generated from large sample sets are susceptible to increased variability because of the day-to-day changes in instrument performance including differences in instrument sensitivity, chromatographic drift of compounds, and varying levels of process contribution or chromatographic carry-over that occur over time. The challenge with this study was also related to the size of the study. Process blank levels differ from run to run as different consumable lots are used over time. Based on the rate of presumed false positives in this study and a closer look at the data it appears that a stricter overall requirement of experimental area counts over process blank area counts is required to account for the more variable process blank levels over time in the future. These issues likely contributed to the increased false positive detection of metformin in this study.

Unlike many biochemicals which can be expected to be found in most, if not all, samples, drugs are inherently present in only a fraction of all the samples in a population study. A careful balance must be struck between the ability to detect drugs (sensitivity), which is required to account for drug dosing and clearance rates, and the avoidance of false positives. Several approaches can be utilized to reduce the false positive rate and increase confidence in the presence of a particular drug. The presence of primary or secondary metabolites of the active substance is a strong indicator that the drug is present, and that the detection is valid in any given sample. Unfortunately, some drugs, including metformin discussed here, do not lend themselves to this form of further scrutiny as it is cleared from the body without modification. Another approach is to require detection of a drug by multiple analytical methods from the same sample. As many untargeted methods employ multiple LC-MS methods for the analysis of each sample the detection of the drug on multiple arms greatly increases the confidence in the presence of the drug. Lastly, a requirement of a high-quality fragmentation spectra match for the potential drug in each sample, rather than in a preponderance of aligned peaks or in the aligned peak in a technical replicate only, provides an elevated level of confidence in the presence of the drug without the need to use other identifications as a part of the analysis. Considering the observed and presumed false positive rate of metformin detection, the data could be re-interrogated using the above noted additional scrutiny to reduce false positives.

In balance with more stringent criteria to reduce false positives is the likely reciprocal increase in false negatives. The clearance rate of drugs coupled with the time from dosing to collection and the dosage-to-weight ratio can mean that the compound falls below the limit of detection of the method, particularly when more stringent criteria are used. Under-reporting of drugs can be mitigated using a robust library built from authentic standards of known drugs, as well as their known metabolites. This increases the ability to detect some drugs for a period after initial dosing as they are metabolized by the body over time.

Finally, an in our opinion unlikely general explanation for our observations, but possibly still relevant to individual cases, is the presence of certain molecules in the drinking water. A very recent global study on the presence of pharmaceutical pollutants in river waters reported numerous of the metabolites that we find under-reported here, including metformin and acetaminophen [25].

In the end, both human and technical factors were at play and will likely continue to play a role in the noted discrepancies between the reporting and detection of drug metabolites (Table 5). As always, in both instances, it is important to identify which factors can be better controlled, such as the human factors of memory and honesty by taking a careful medical history and accurate collation of the self-reporting data, and for the technical issues the careful analysis and interpretation of the obtained data and further improvement of detection and metabolite annotation methods. Large biobank studies can play a major role in this process.

## 4. Materials and Methods

### 4.1. Qatar Biobank

QBB is a population study including Qatari nationals and long-term residents (≥15 years living in Qatar) aged 18 years and above [9,19]. Information on drug use was collected using a nurse-administered questionnaire using the formulation “*Are you taking any over-the-counter medication or prescription medicines regularly? For example, daily, weekly, monthly or every few months—such as depot injections?*” (QBB item NQ_A25). Up to 30 free text entries were allowed.

### 4.2. Drugbank and ATC Annotation

The vocabulary of DrugBank Release Version 5.1.6, which is released under a Creative Common’s CC0 International License, was downloaded (https://www.drugbank.ca/releases/latest#open-data, accessed on 25 April 2020). ATC annotation was retrieved using the DrugBank annotation.

### 4.3. Non-Targeted Metabolomics Measurements

In total, 3000 EDTA blood plasma aliquots were analyzed using a non-targeted metabolomics platform from that has been implemented by Metabolon Inc. (Morrisville, NC, USA) at the Anti-Doping Laboratory–Qatar (ADLQ) in collaboration with Weill Cornell Medicine–Qatar (WCM-Q), Hamad Medical Corporation (HMC), and Qatar Biomedical Research Institute (QBRI). Instrumentation, protocols, quality control processes and metabolite annotations were identical to those implemented in Durham and previously described [26]. Briefly, samples were prepared using the automated MicroLab STAR system from Hamilton Company. Several recovery standards were added prior to the first step in the extraction process for QC purposes. To remove protein, dissociate small molecules bound to protein or trapped in the precipitated protein matrix, and to recover chemically diverse metabolites, proteins were precipitated with methanol under vigorous shaking for 2 min (Glen Mills GenoGrinder 2000) followed by centrifugation. The resulting extract was divided into five fractions: four for analysis and one sample was reserved for backup. Samples were placed briefly on a TurboVap (Zymark) to remove the organic solvent. The sample extracts were stored overnight under nitrogen before preparation for analysis.

Three LC-MS systems, consisting of a Waters ACQUITY ultra-performance liquid chromatography (UPLC) unit and a Thermo Scientific Q-Exactive high resolution/accurate mass spectrometer, interfaced with a heated electrospray ionization (HESI-II) source and Orbitrap mass analyzer operated at 35,000 mass resolution were used. One LC/MS system used acidic positive ion conditions with a C18 column and ran two methods, one chromatographically optimized for more hydrophilic compounds and the other for more hydrophobic compounds. The second LC/MS system used a method optimized for basic negative ions and a separate dedicated C18 column. The third LS/MS system used negative ionization following elution from a HILIC column. Details of the methods are described in the QC report provided by Metabolon (Appendix A).

All measurements were performed on a fee-for-service basis, ordered and funded by the Qatar Biobank (QBB). In total, 1018 samples of a non-related clinical study (not analyzed here) were run together with the 3000 samples of QBB. The 4018 samples were randomized onto 112 run days with a capacity of 36 samples per day, accounting for sex, age, body mass index, diabetes state, prevalent hypertension, and HbA1c% in such a way that none of these parameters associated with run day. Raw data were transferred to Metabolon and annotated using their proprietary in-house software. Briefly, Metabolon’s biochemical identifications is based on three criteria: retention index within a narrow RI window of the proposed identification, accurate mass match to the library +/− 10 ppm, and the MS/MS forward and reverse scores between the experimental data and authentic standards. The MS/MS scores are based on a comparison of the ions present in the experimental spectrum to the ions present in the library spectrum. More than 3300 commercially available purified standard compounds have been acquired and registered into LIMS for analysis on all platforms for determination of their analytical characteristics. Additional mass spectral entries have been created for structurally unnamed biochemicals, which have been identified by virtue of their recurrent nature (both chromatographic and mass spectral). Library matches for each compound were checked for each sample and corrected if necessary.

Cases of limited confidence in the biochemical annotation or existence of multiple isomers is indicated as follows: biochemical name followed by ‘*’ indicates a compound that has not been officially confirmed based on a standard, but that Metabolon is confident in its identity. Biochemical name followed by ‘**’ indicates a compound for which a standard is not available, but Metabolon is reasonably confident in its identity, or the information provided. Biochemical name followed by ‘(#)’, where # is a number, indicates a compound that is a structural isomer of another compound in the Metabolon spectral library. For example, a steroid that may be sulfated at one of several positions that are indistinguishable by the mass spectrometry data or a diacylglycerol for which more than one stereospecific molecule exists.

Peaks were quantified using area-under-the-curve. Data normalization was performed in run-day blocks by registering the medians to equal one and normalizing each data point proportionately.

Instrument variability was determined by calculating the median relative standard deviation (RSD) for the internal standards that were added to each sample prior to injection into the mass spectrometers. The overall RSD for instrument variability based on internal standards was 12% and the total process variability determined using endogenous metabolites detected in reference sample was 16% and met Metabolon’s acceptance criteria. The provided dataset comprised a total of 1159 biochemicals, 937 compounds of known identity (named biochemicals) and 222 unidentified compounds, marked by “X-”, followed by a numeric identifier and designated “unknown” in this manuscript.

### 4.4. Peak Detection Criteria

For a peak to be “detected”, it must meet defined criteria for signal to noise ratio, minimum peak area, mass tolerance, and peak width by a proprietary integration and peak picking algorithm. Peaks meeting these criteria are subjected to further review addressing parameters such as peak area above process (water) blanks, consistency of chromatographic retention, and overall peak shape. Any inconsistencies in peak peaking including baselining, peak splitting and peak integration inconsistencies are also corrected during this QC stage. At this stage, the peak has been both detected, but also approved based on meeting defined QC parameters. In the instance of this manuscript, the dominant factor in whether a peak was both detected and approved was related to assessment of the peak areas in experimental samples to the peak areas within the process blanks (diH2O taken through the entire process). This analysis is in place to allow the removal of any artifacts related to the preparation of the samples such as artifacts that may arise from lab consumables or solvents and include plasticizers and releasing agents. Any experimental peak with a peak area of less than 3X the process blanks is excluded/rejected from the analyses.

### 4.5. UK Biobank Medication

UK Biobank medication data (Appendix A) was downloaded form a publicly available website (https://biobank.ndph.ox.ac.uk/showcase/field.cgi?id=20003, accessed on 27 April 2020, counts of participants/items last updated 10 January 2020). This data set contains 1,380,303 items of data covering 372,854 participants. 3735 different medications were reported to have been taken by at least one UKB participant, most frequently mentioned was paracetamol (102,058), aspirin (72,926), ibuprofen (67,388), and simvastatin (64,358). Details on the assessment can be found on the UK Biobank web page for category 100075 (Medications—Verbal interview—UK Biobank Assessment Centre). We used information from Supplementary Table S1 of [13] to link medication-use items in UK Biobank to drug names.

### 4.6. Statistical Analysis

All statistical analyses were conducted using R (version 4.1.0 and above, https://cran.r-project.org/bin/windows/base/old/4.1.0/ accessed on 13 February 2022) and Rstudio (version 1.4.1717 and above, https://docs.rstudio.com/ide/server-pro/1.4.1717-2/r-versions-1.html accessed on 13 February 2022).

## Figures and Tables

**Figure 1 metabolites-12-00249-f001:**
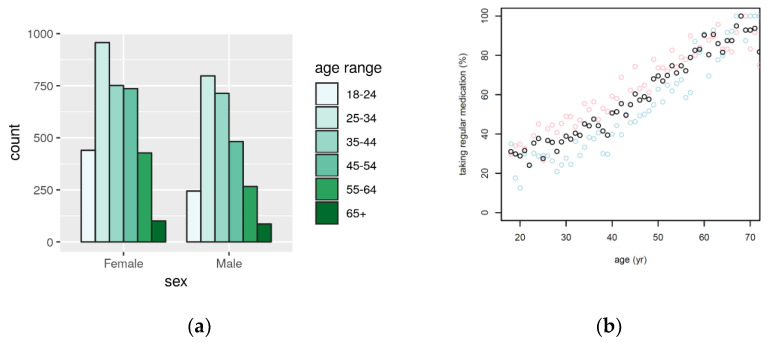
Histogram of the 6000 QBB participants included in this study, presented by sex and age (**a**); fraction of male (blue), female (pink), and all (black) QBB participants who self-reported using at least one over-the-counter or prescription drug, stratified by age (**b**).

**Figure 2 metabolites-12-00249-f002:**
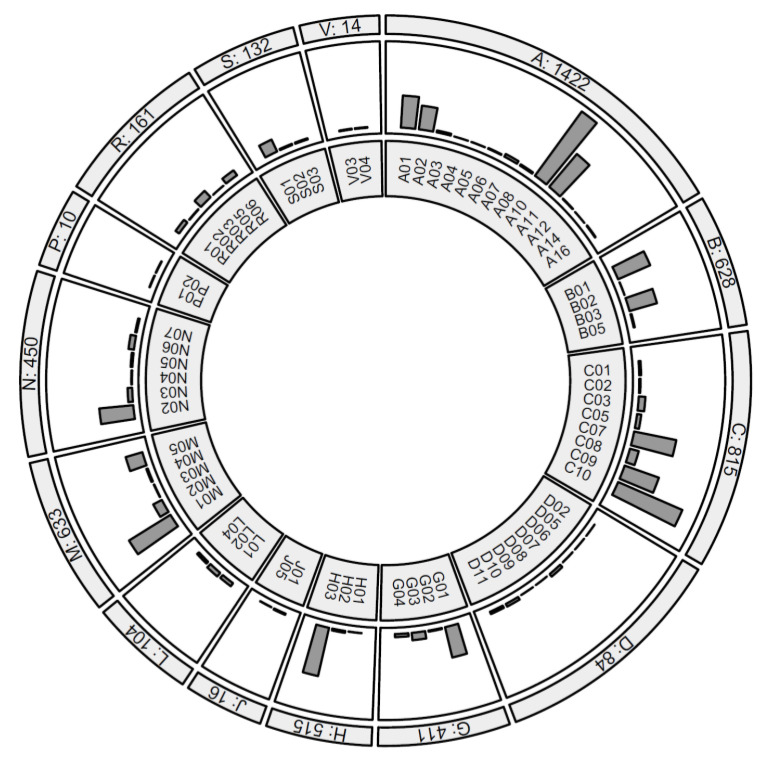
Counts of drugs taken by 6000 QBB participants, ordered by ATC therapeutic subgroup. Subgroups with over 100 counts are: A10:DRUGS USED IN DIABETES (795), C10:LIPID MODIFYING AGENTS (700), H03:THYROID THERAPY (495), M01:ANTIINFLAMMATORY AND ANTIRHEUMATIC PRODUCTS (488), A11:VITAMINS (430), C07:BETA BLOCKING AGENTS (430), C09:AGENTS ACTING ON THE RENIN-ANGIOTENSIN SYSTEM (363), B01:ANTITHROMBOTIC AGENTS (350), N02:ANALGESICS (341), A01:STOMATOLOGICAL PREPARATIONS (321), G01:GYNECOLOGICAL ANTIINFECTIVES AND ANTISEPTICS (312), B03:ANTIANEMIC PREPARATIONS (285), A02:DRUGS FOR ACID RELATED DISORDERS (239), M05:DRUGS FOR TREATMENT OF BONE DISEASES (174), S01:OPHTHALMOLOGICALS (132). See Appendix A for all subgroup definitions and all counts.

**Figure 3 metabolites-12-00249-f003:**
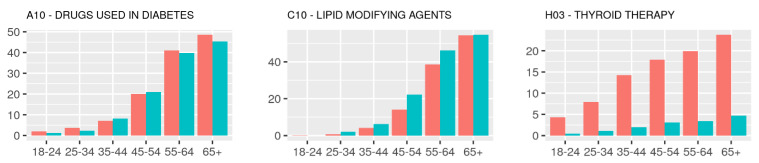
Frequencies (by age and sex group) of the twelve most frequently reported drug classes (ATC code level 2) by age and sex; percentages of males (cyan) and females (red) who reported at least one drug of the indicated classes in their respective age group, computed based on the responses of 6000 participants of QBB.

**Figure 4 metabolites-12-00249-f004:**
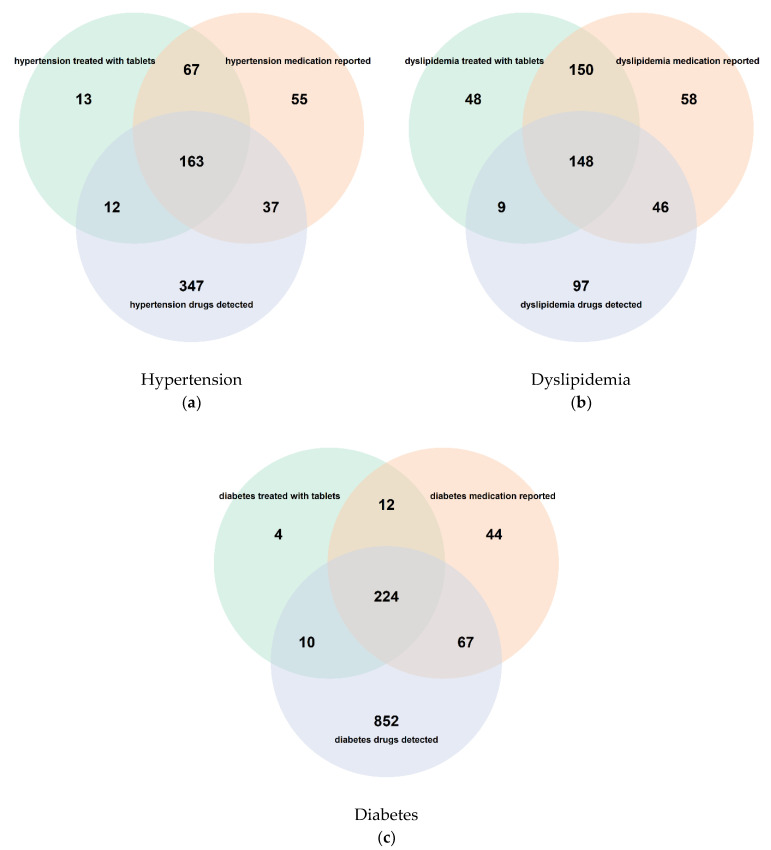
Overlap between individuals who gave an affirmative answer to the question of whether they were treated for a given disorder with tablets (green), who self-reported the use of at least one medication with an indication for the given disorder (orange), and who provided a blood sample with at least one metabolite of a drug with an indication for the given disorder detected (blue); the given disorders were hypertension (**a**), dyslipidemia (**b**), and diabetes (**c**).

**Figure 5 metabolites-12-00249-f005:**
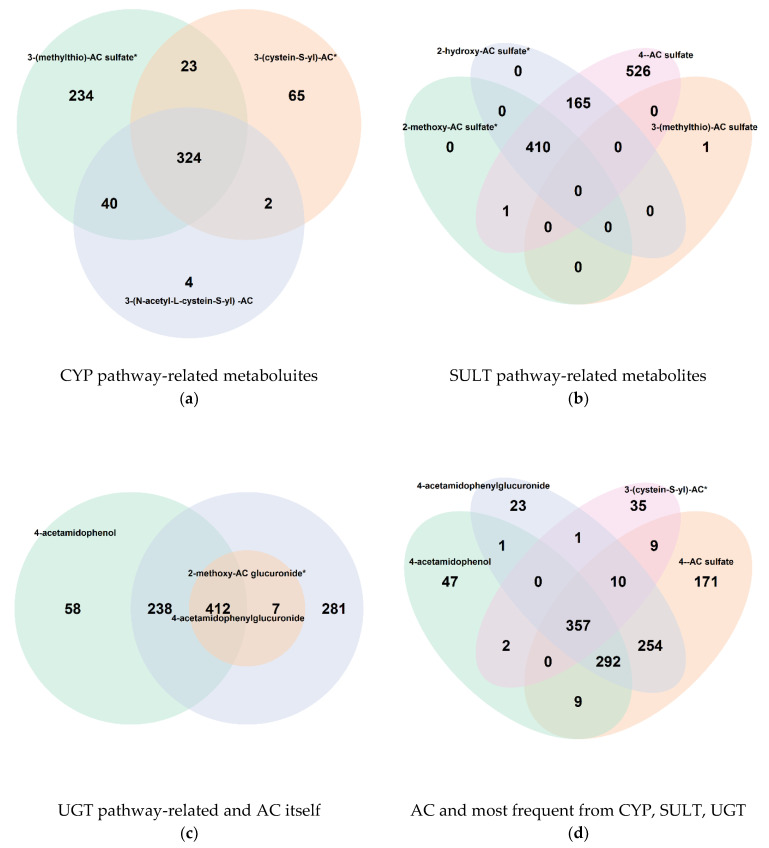
Overlap between acetaminophen metabolites detected in blood. Venn diagrams for the nine acetaminophen (AC) metabolites detected in this study, grouped by metabolites specific to the CYP, SULT, and UGT pathways, resp. (**a**–**c**), and for the most frequently detected metabolite from each pathway (**d**); all nine AC metabolites were detected together in 310 (11.0%) samples, and at least one AC metabolite was detected in 1255 (44.7%) samples.

**Figure 6 metabolites-12-00249-f006:**
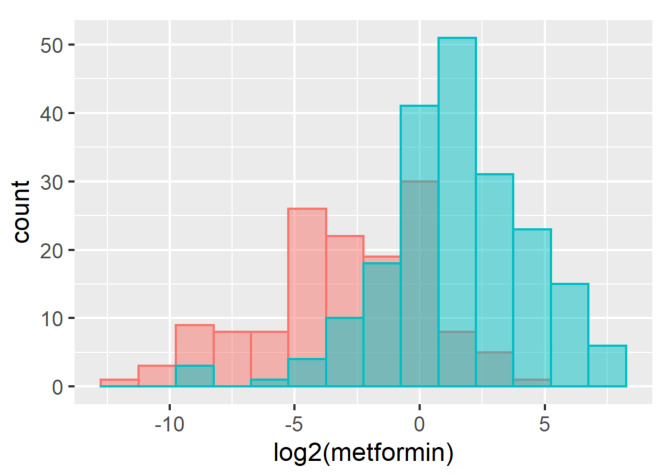
Histogram of metformin counts (batch-corrected, log-scaled, relative units) in individuals who self-reported using metformin (blue) and individuals who were unlikely to use metformin (red); Of 571 individuals who are unlikely to take metformin, 431 (75.5%) have no metformin detected in their blood samples, whereas 140 (24.5%) do; of the 220 individuals who self-report using metformin, 203 (92.3%) have metformin detected in their blood and only 17 (7.7%) do not.

**Table 1 metabolites-12-00249-t001:** Demographics of the 6000 QBB participants who were included in this study. Except for HbA1c and BMI, all information is self-reported using questionnaires dispensed at baseline.

Variable	Value(s)
Mean age in years (s.d.)	39.7 (12.8)
Female (%)	3412 (56.9%)
Mean BMI in kg/m^2^ (s.d.)	28.9 (6.2)
Mean HbA1c % (s.d.)	5.7 (1.26)
Diagnosed with diabetes (%)	1041 (17.4%)
Treated for diabetes with tablets (%)	717 (12.0%)
Treated for diabetes with insulin (%)	286 (4.8%)
Treated for high cholesterol (%)	1829 (30.5%)
Treated for high cholesterol with tablets (%)	815 (13.6%)
Treated for high blood pressure (%)	966 (16.1%)
Treated for high blood pressure with tablets (%)	652 (10.9%)
Regular smoker (%)	602 (10.0%)
Reported at least one over-the-counter or prescription drug (%)	3086 (51.4%)

**Table 2 metabolites-12-00249-t002:** Number of self-reported medication by ATC anatomical main group, based on responses of 6000 QBB participants, whereof 3086 reported to take at least one kind of medication.

ID	Anatomical Main Group	N ^1^
A ^1^	ALIMENTARY TRACT AND METABOLISM	1422
B	BLOOD AND BLOOD FORMING ORGANS	628
C	CARDIOVASCULAR SYSTEM	815
D	DERMATOLOGICALS	84
G	GENITO URINARY SYSTEM AND SEX HORMONES	411
H	SYSTEMIC HORMONAL PREPARATIONS, EXCL. SEX HORMONES AND INSULINS	515
J	ANTIINFECTIVES FOR SYSTEMIC USE	16
L	ANTINEOPLASTIC AND IMMUNOMODULATING AGENTS	104
M	MUSCULO-SKELETAL SYSTEM	633
N	NERVOUS SYSTEM	450
P	ANTIPARASITIC PRODUCTS, INSECTICIDES AND REPELLENTS	10
R	RESPIRATORY SYSTEM	161
S	SENSORY ORGANS	132
V	VARIOUS	14

^1^ Note that an active drug molecule can have multiple ATC codes assigned to it.

**Table 3 metabolites-12-00249-t003:** Self-reported medication (N self) and drug metabolites detected in blood (N blood); The total number of samples in this analysis was N = 2807. Included are all mutual best-hits with Bonferroni-significant associations (Fisher test, *p*-value < 0.05/N-self-reported/N-detected = 0.05/394/119 = 1.1 × 10^−6^). All pairwise associations that reached nominal significance (*p* < 0.05) are in Appendix A.

Self-Reported Medication	N Self	Metabolite Detected in Blood	N Blood	*p*-Value (Fisher)	Self Only	Blood Only	Both
metformin	220	metformin	846	8.7 × 10^−92^	17	643	203
atorvastatin	90	o-hydroxyatorvastatin	110	3.1 × 10^−76^	27	47	63
valsartan	40	valsartan	171	7.0 × 10^−46^	2	133	38
losartan	24	losartan	27	3.7 × 10^−38^	5	8	19
gliclazide	47	gliclazide	468	6.2 × 10^−30^	4	425	43
indapamide	15	indapamide	34	9.2 × 10^−25^	2	21	13
warfarin	9	10-hydroxywarfarin	13	2.4 × 10^−23^	0	4	9
atenolol	18	atenolol	140	4.8 × 10^−22^	1	123	17
acetylsalicylic acid	111	salicyluric glucuronide	1292	2.5 × 10^−21^	13	1194	98
escitalopram	10	citalopram/escitalopram	31	5.4 × 10^−21^	0	21	10
esomeprazole	26	omeprazole	67	7.4 × 10^−21^	10	51	16
glimepiride	18	glimepiride	25	1.7 × 10^−17^	8	15	10
topiramate	7	topiramate	20	2.9 × 10^−16^	0	13	7
pantoprazole	12	pantoprazole	88	4.0 × 10^−12^	3	79	9
celecoxib	18	celecoxib	58	1.5 × 10^−11^	9	49	9
pregabalin	7	pregabalin	16	6.3 × 10^−11^	2	11	5
fexofenadine	12	fexofenadine	57	7.2 × 10^−10^	5	50	7
perindopril	12	perindopril	14	1.1 × 10^−9^	7	9	5
pioglitazone	5	hydroxypioglitazone	23	1.7 × 10^−8^	1	19	4
acetaminophen	17	2-methoxyacetaminophen glucuronide	419	2.1 × 10^−8^	4	406	13
montelukast	5	montelukast	26	2.9 × 10^−8^	1	22	4
repaglinide	2	repaglinide	3	7.6 × 10^−7^	0	1	2

**Table 4 metabolites-12-00249-t004:** Nine metabolites of the drug paracetamol (acetaminophen) were individually detected in between 370 and 1102 of the 2807 blood samples, with a joint detection of all nine metabolites in 310 samples. LC/MS characteristics of the detected molecules are provided together with the biochemical detoxification pathway in which the respective molecules are involved (see text).

Biochemical	N Detected (% of 2807)	N Overlap (% of 17)	LC/MS Mode	Retention Index	Mass	Pathway
4-acetaminophen sulfate	1102 (39.3%)	16 (98.5%)	Neg	1792	230.01287	SULT
4-acetamidophenyl glucuronide	938 (33.4%)	16 (98.2%)	Neg	1400	326.08814	UGT
4-acetamidophenol	708 (25.2%)	15 (97.6%)	Neg	2173.7	150.05605	Paracetamol
3-(methylthio) acetaminophen sulfate *	621 (22.1%)	14 (97.3%)	Neg	2265	276.00059	CYP/SULT
2-hydroxyacetaminophen sulfate *	575 (20.5%)	14 (97.1%)	Neg	1674	246.00778	SULT
2-methoxyacetaminophen glucuronide *	419 (14.9%)	13 (96.0%)	Neg	1633	356.0987	UGT
2-methoxyacetaminophen sulfate *	411 (14.6%)	12 (95.9%)	Neg	1949	260.02343	SULT
3-(N-acetyl-L-cystein-S-yl) acetaminophen	370 (13.2%)	11 (95.5%)	Neg	2094	311.07072	CYP
3-(cystein-S-yl) acetaminophen *	414 (14.7%)	11 (96.0%)	Pos Early	2420	271.07471	CYP

Biochemical name followed by ‘*’ indicates a com-pound that has not been officially confirmed based on a standard, but that Metabolon is confident in its identity.

**Table 5 metabolites-12-00249-t005:** Possible reasons for the apparent under- and over-detection of drug metabolites.

Drug Metabolite	Possible Reason	Example
Non- identification	The metabolite is not in the platform library	Metabolites of unknown identity, X – 17348 with esomeprazole
Non-detection	The metabolite is not captured by the current LC/MS protocol	Rosuvastatin, reported by 141/6000 participants, but not detected
Under-detection	Blood levels below level of detection (metabolite half-life, timing of sampling)	Atorvastatin
Over-detection	Medication is taken, but not reported	Painkillers, anti-depressants
Over-detection	Carry-over from other samples	Not observed
Over-detection	Misidentification of other molecules (overlapping signals)	Metformin
Ubiquitous	Metabolite is endogenous, supplementation by medication cannot be determined by presence/absence of detection	Levothyroxine, reported by 492/6000 participants

## Data Availability

Access to Qatar Biobank data can be obtained through an established ISO-certified process by submitting a project request at https://www.qatarbiobank.org.qa/research/how-apply which is subject to approval by the Qatar Biobank IRB committee.

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
