# Peer review of "Matching Drug Metabolites from Non-Targeted Metabolomics to Self-Reported Medication in the Qatar Biobank Study"

_metabolites, 2022, doi:10.3390/metabo12030249_

Round 1

Reviewer 1 Report

This is an interesting study on a large population. Overall the manuscript is clear and well written, although the results part could be condensed a bit. I have one major concern and other minor comments as listed below.

Major comments

There is one major concern related to the laboratory analysis that potentially has a major effect on the results and discussion. The work relies on the ability of the methods to “detect” the presence or absence of a drug or its metabolite, but the definition of “detection” is not given. The results include significant numbers of “detections” for drugs or their metabolites the study participants did not report taking (such as Figure 4), and this suggests presence of false positives. This is was noted and some reasons were discussed; carry-over given as one potential cause (line 379-382). Carry-over was not considered a cause of the issue, although no results were shown to support this conclusion. However, this is a critical aspect of the study and needs to be addressed. Like in doping control, the risk of false positives must be avoided in a study like this, so there needs to be a clear definition of “detection”. This is now entirely missing, and I believe carry-over is in fact potentially contributing to the high number of “over-detections” as this is a well-known phenomenon in LC-MS metabolomics and difficult to completely avoid. Other sources of false positivels might be a lack of reporting thresholds based on abundance, S/N ratio, or some other metrics set for the chromatographic peaks, or specificity issues with the annotation. In any case, it is necessary to define “detection” and employ suitable reporting thresholds to the data. This may improve the comparability of the self-reported and metabolomics-based data and lessen the need to discuss possible reasons for the mismatch.

Other comments

Line 23: “self-reported drugs being detected” consider drugs and their metabolites

Line 25: “Possible explanations include under-reporting of 25 over-the-counter medications from the study participants,…” consider adding “or inability of the methods to detect them,…

Line 26: “over-annotation of low abundance metabolites” this is not clear, please revise

Results, Demographics: Many of the details written are already in the Table and Figure, no need to repeat in the text.

Table 1: “Medication” needs to be somehow defined

Figure 1: “taking” and “taking regular medication” are vague terms, better definition of the use of medication would help.  

Line 116: Better define what is meant by “active molecule” here and later in text? Thinking of prodrugs, or even acetylsalicylic acid/salicylic acid.

Line 122: Why are vitamins and supplements are included in a study on drugs/medications? I strongly suggest excluding these, unless you change the title and introduction so that it is clear this study covers all of them and not only drugs.

Figure2: It is difficult to see this as an important figure that supports the main results. I suggest moving this to supplements

Figure 3: Add axis titles. Titles on (d) and (g) are truncated.

Section 2.3. It is distracting to read a section on purely self-reported results from UK Biobank, in a study that is titled “Matching drug metabolites from non-targeted metabolomics to 2 self-reported medication in the Qatar Biobank study” I suggest removing this section, to keep the focus on the QBB and the metabolomics-self reporting comparison. Comparing self-reported data between QBB and UK Biobank does not seem too important for this paper.

Line 210: “Detection” needs a clear definition.

Line 227: “biochemically closely related” This is not clear. Do you mean metabolites and their precursors?

Line 231: “biochemical matches” Not clear what this means

Line 233-245: I suggest leaving this out. This part is a bit too speculative and focuses on unknown compounds without clear outcome.

Line 246-258: The results in this part strongly suggest presence of false positive annotations. See my major comments.

Section 2.5: avoid repeating results that are in the figures 3 and 4.

Line 353: What is “active molecule”? See line 116.

Line 356: What do the asterisks signify?

Line 357-359: This is too speculative. It is also not impossible that at least some of these are false positives. Overall, I suggest leaving UK Biobank out form the study.

Line 377-382: This suggest high likelihood of false positive detections. It is critical to assess carry-over and/or use sufficiently high reporting thresholds. If needed as a proof, data should be shown.  

Author Response

Please see the attachment for  the point-by-point response to all reviewer’s comments.

Reviewer 2 Report

1. When author categoried the data, gender, age, BMI, and so on were considered. However, the difference of racies was not described. If it is possible to add the races, the result of comparison or statistics can be able to be more accurate. How about this opinion?

2. In the line 515, the author said the metabolon criteria. It would be better if you describe the metabolon criteria in detail.

3. The author used HbAC1 as a factor. Is there any reason to use this?

4. When the analysis of LC-MS/MS, the carry over can be happened during the analysis, especially, the sample after analysis of high concentration of sample. How did you overcome this carry-over or how did you confirm that the carry-over did not effect on?

5. During the circulation via liver in the body, the drug can be changed the metabolite and be changed the ratio of drug to the metabolite. When the author analysis the blood samples, how did you set the sampling time or compensate the result if you did not set to the same sampling time point.

6. When the analysis using LC-MS/MS or prepartion of samples, the unstable drug and drug metabolite can be degradated or changed to other form by the enzyme. How did the author control the samples to be stable.

7. The preparation method was not described in this manuscript. If possible, even though the analysis was performed in the CRO, please the athour described the method at least briefly.

Author Response

(The authors gave the same response as above.)

Reviewer 3 Report

This manuscript present a study on a bank of plasma from Quatar patients on the possibility to retrieve use of medications by metabolomics (HPLC/MS) and comparison with a questionnaire submitted to each patient on disease and drug intake.

The study is well written and interesting.

It is strange that so many drugs or drug metabolites are found although not reported by the patients. It seems that this problem was also present in a similar UK study with UK patients.

I noted a few problems that could be easily corrected:

In fig 3h, it is starnge that male papteint take ginecological antiseptic or antibiotics. Is this class of drug well named?

Page 10 Table 3. I think you should better explain the meaning of N in each column.

Page 13 : you should give more details on which ion you monitor for metformin metabolites. It is unlikely that a compound with the correct retention time and the correct accurate mass (within 10 ppm) for the metabolite be a different compound. But may be the selected metabolite could also be produced from other starting endogenous compound or food components. Have you though about that? Could you discuss it?

Since you do not give the list of mass ion monitored it is difficult to appreciate.

Page 16 : Reference 25 refers to either a LTQ intrument or a LTQ-FT-Cyclotronic intrument. In the current paper you are using a Q-Exactive HF. Much more practical and quite accurate. But this should be explained.

I find the paper really interesting. The supplementary table giving a list of compounds as Xcel table is interesting. Is it possible to give some more information on the HPLC/MS methodology for the compounds cited in the text: HPLC system, Retention time, MS ion, MS/MS ion monitored.

And may be a scheme on metformin and metformin metabolites studied as an example?

Can you explain why the rosuvastatin metabolites were not found? Do you find them when you check for plasma of a volunteer taking rosuvastatin?

Once these question have been answered, I thing that the paper should be acceptable

Author Response

(The authors gave the same response as above.)

Round 2

Reviewer 2 Report

All comment that I indicated were corrected in a revised manuscript.